# Variational Stochastic Gradient Descent for Deep Neural Networks

**Haotian Chen**[*]                                                                            *chenhaotian.jtt@gmail.com*
*Department of Mathematics and Computer Science,*
*Eindhoven University of Technology, Netherlands.*

**Anna Kuzina**[*]                                                                             *anna.kzna@gmail.com*
*Department of Computer Science,*
*Vrije Universiteit Amsterdam, Netherlands.*

**Babak Esmaeili**                                                                            *b.esmaeili@tue.nl*
*Department of Mathematics and Computer Science,*
*Eindhoven University of Technology, Netherlands.*

**Jakub M. Tomczak**                                                                          *j.m.tomczak@tue.nl*
*Department of Mathematics and Computer Science,*
*Eindhoven University of Technology, Netherlands.*

**Reviewed on OpenReview:** *https://openreview.net/forum?id=xu4ATNjcdy*

## Abstract

Optimizing deep neural networks is one of the main tasks in successful deep learning. Current state-of-the-art optimizers are adaptive gradient-based optimization methods such as ADAM. Recently, there has been an increasing interest in formulating gradient-based optimizers in a probabilistic framework for better modeling the uncertainty of the gradients. Here, we propose to combine both approaches, resulting in the Variational Stochastic Gradient Descent (VSGD) optimizer. We model gradient updates as a probabilistic model and utilize stochastic variational inference (SVI) to derive an efficient and effective update rule. Further, we show how our VSGD method relates to other adaptive gradient-based optimizers like ADAM. Lastly, we carry out experiments on two image classification datasets and four deep neural network architectures, where we show that VSGD outperforms ADAM and SGD.

## 1 Introduction

The driving force for deep learning success is efficient and effective optimization (Bottou, 2012; Sun, 2020). Deep Neural Networks (DNNs) introduce multiple optimization challenges due to their complexity, size, and loss landscape with multiple local minima, plateaus, and saddle points. Since the first attempt to train DNNs with stochastic gradient descent (SGD), there have been multiple approaches to speed up the learning process and improve the final performance of DNNs.

Nowadays, adaptive gradient-based methods are the leading optimizers in deep learning. The first advancements arrived with the idea of adding momentum (Polyak, 1964) to the SGD update rule (SGDM) which demostrated impressive performance in training DNNs (Sutskever et al., 2013). The next major breakthrough was ADAM (Kingma & Ba, 2015), which uses the first and second momenta to adapt the learning rate and gradients. Currently, ADAM is probably the most widely used optimizer in deep learning due to its relative insensitivity to hyperparameter values and much faster initial progress in training compared to SGD (Sun, 2020).

---

* Equal contribution

Recently, there has been an increasing interest in formulating SGD in a probabilistic framework. An example is Mandt et al. (2017) in which SGD is used as an approximate Bayesian inference algorithm analyzed by relating it to the Ornstein-Uhlbeck process. A different approach utilizes a Bayesian perspective on SGD to model the uncertainty for datastreams (Liu et al., 2024).

In this paper, we propose to combine these two approaches, and, as a result, we obtain a probabilistic framework for adaptive gradient-based optimizers. First, we formulate a probabilistic model of SGD. Second, we employ stochastic variational inference SVI (Hoffman et al., 2013) to derive an efficient adaptive gradient-based update rule. We refer to our approach as Variational Stochastic Gradient Descent (VSGD). We apply our method to optimize overparameterized DNNs on image classification. Compared to ADAM and SGD, we obtain very promising results.

Our contributions can be summarized as follows:

1. We propose Variational Stochastic Gradient Descent (VSGD), a novel optimizer that adopts a probabilistic approach. In VSGD, we model the true gradient and the noisy gradient as latent and observed random variables, respectively, within a probabilistic model. Additionally, this unique perspective allows us to manage gradient noise more effectively by adopting distinct noise models for the true and noisy gradients, enhancing the optimizer's robustness and efficiency.

2. We draw connections between VSGD and several established non-probabilistic optimizers, including NORMALIZED-SGD, ADAM, and SGDM. Our analysis reveals that many of these methods can be viewed as specific instances or simplified adaptations of VSGD when a constant noise model is assumed for both the true and observed gradients.

3. Lastly, we carry out an empirical evaluation of VSGD by comparing its performance against the most popular optimizers, namely ADAM and SGD in the context of large-scale image classification tasks using overparameterized deep neural networks. Our results indicate that VSGD not only achieves lower generalization errors, but also converges at a competative rate compared to traditional methods such as ADAM and SGD. We believe that these findings underscore the practical advantages of VSGD, making it a compelling choice for complex deep learning challenges.

## 2 Background knowledge

**Training Neural Nets with Gradient Descent**  Let us consider a supervised setup for deep neural networks (DNNs) in which a DNN with weights $\theta \in \mathbb{R}^D$ predicts a target variable $y \in \mathcal{Y}$ for a given object $\mathbf{x} \in \mathcal{X}$, $f_\theta : \mathcal{X} \to \mathcal{Y}$. Here, we treat $\theta$ as a vector to keep the notation uncluttered. Theoretically, finding *best* values of $\theta$ can be achieved by *risk minimization*: $\mathcal{R}(\theta) = \mathbb{E}_{p(\mathbf{x},y)}[\ell(f_\theta(\mathbf{x}), y)]$, where $\ell$ is a loss function. In practice, we do not have access to $p(\mathbf{x}, y)$, but we are given a dataset $\mathcal{D}$. Then, training of $f_\theta$ corresponds to *empirical risk minimization* (Bottou, 2012; Sun, 2020): $\mathcal{L}(\theta; \mathcal{D}) = \frac{1}{N} \sum_{n=1}^{N} \ell(f_\theta(\mathbf{x}_n), y_n)$. We assume that the loss function is $k$-differentiable, $\ell \in \mathcal{C}^k$, at least $k = 1$.

Many optimization algorithms for DNNs are based on *gradient descent* (GD). Due to computational restrictions on calculating the full gradient (that is, over all training datapoints) for a DNN, the stochastic version of GD, *stochastic gradient descent* (SGD), is employed. SGD results in the following update rule:

$$\theta_t = \theta_{t-1} - \eta_t \hat{g}_t, \tag{1}$$

where $\eta_t$ is a learning rate, $\hat{g}_t = \nabla_\theta \mathcal{L}(\theta; M)$ is a noisy version of the full gradient calculated over a mini-batch of $M$ data points, $\mathcal{L}(\theta; M) = \frac{1}{|M|} \sum_{n \in M} \ell(f_\theta(\mathbf{x}_n), y_n)$, and we assume that $\mathbb{E}[\hat{g}_t] = g_t$.

**Related work**  Optimization of DNNs is a crucial component in deep learning research and, especially, in practical applications. In fact, the success of deep learning models is greatly dependent on the effectiveness of optimizers. The loss landscapes of DNNs typically suffer from multiple local minima, plateaus, and saddle points (Du & Lee, 2018), making the optimization process challenging.

Since the introduction of backpropagation (Rumelhart et al., 1986; Schmidhuber, 2015), which is based on SGD updates, multiple techniques have been proposed to improve SGD. Some improvements correspond to a better initialization of layers that results in a *warm start* of an optimizer (e.g., (Glorot & Bengio, 2010; He et al., 2015; Sutskever et al., 2013)), or pre-defined learning schedules (e.g., (Loshchilov & Hutter, 2016)).

Another active and important line of research focuses on adaptive gradient-based optimizers. Multiple methods are proposed to calculate adaptive learning rates for DNN, such as RMSprop (Graves, 2013; Tieleman & Hinton, 2012), AdaDelta (Zeiler, 2012), and Adam (Kingma & Ba, 2015). Adam is probably the most popular optimizer due to its relative insensitivity to hyperparameter values and rapid initial progress during training (Keskar & Socher, 2017; Sun, 2020). However, it was pointed out that Adam may not converge (Reddi et al., 2018), leading to an improvement of Adam called ADMSGrad (Reddi et al., 2018). There are multiple attempts to understand Adam that lead to new theoretical results and new variants of Adam (e.g., (Barakat & Bianchi, 2019; Zou et al., 2019)). In this paper, we propose a new probabilistic framework that treats gradients as random variables and utilizes stochastic variational inference (SVI) (Hoffman et al., 2013) to derive an estimate of a gradient. We indicate that our method is closely related to Adam and is an adaptive gradient-based optimizer.

A different line of research focuses on $2^{nd}$-order optimization methods that require the calculation of Hessian matrices. In the case of deep learning, it is infeasible to calculate Hessian matrices for all layers. Therefore, many researchers focus on approximated $2^{nd}$-order optimizers. One of the most popular methods is KFAC (Martens & Grosse, 2015). Recently, there have been new developments in this direction (e.g., (Eschenhagen et al., 2023; Lin et al., 2021; 2023)). However, extending K-FAC introduces memory overhead that could be prohibitive, and numerical instabilities might arise in low-precision settings due to the need for matrix inversions or decompositions. Here, we focus on the $1^{st}$-order optimizer. However, we see great potential for extending our framework to the $2^{nd}$-order optimization.

There are various attempts to treat SGD as a probabilistic framework. For instance, Mandt et al. (2017) presented an interesting perspective on how SGD can be used for approximate Bayesian posterior inference. Moreover, they showed how SGD relates to a stochastic differential equation (SDE), specifically, the Ornstein-Uhlbeck process. The idea of perceiving SGD as an SDE was further discussed in, e.g., (Yokoi & Sato, 2019) with interesting connections to stochastic gradient Langevin dynamics (Welling & Teh, 2011). Another perspective was on the phrasing SGD as a Bayesian method either for sampling (Mingard et al., 2021), modeling parameter uncertainty for streaming data (Liu et al., 2024) or generalization (Smith & Le, 2017). A different approach uses linear Gaussian models to formulate the generation process of the noisy gradients and infer the hidden true gradient with the Kalman filter or Bayesian filter, e.g. (Bittner & Pronzato, 2004; Vuckovic, 2018; Yang, 2021). In this paper, we propose to model dependencies between *true* gradients and *noisy* gradients through the Bayesian perspective using a novel framework. Our framework treats both quantities as random variables and allows efficient inference and incorporation of prior knowledge into the optimization process. A similar line of work is the natural gradient-based methods for Bayesian Inference, where the optimizer uses natural gradients instead of Euclidean gradients (Khan et al., 2018; Osawa et al., 2019). Our work is different from this line of work, as they apply Adam to learn the variational posteriors of the model. VSGD, on the other hand, learns a variational posterior for the gradients themselves and may not necessarily be applied to estimate the posterior of the model parameters.

## 3 Methodology

### 3.1 Our approach: VSGD

**A probabilistic model of SGD** There are several known issues with applying SGD to the training of DNNs (Bottou, 2012; Sun, 2020), such as slow convergence due to gradient explosion/vanishing or too noisy gradient, poor initialization of weights, and the need for a hyperparameter search for each architecture and dataset. Here, we would like to focus on finding a better estimate of the *true* gradient. To achieve that, we propose to perceive SGD through a Bayesian perspective, in which we treat noisy gradients as observed variables and want to infer the posterior distributions over true gradients. Subsequently, we can then use the

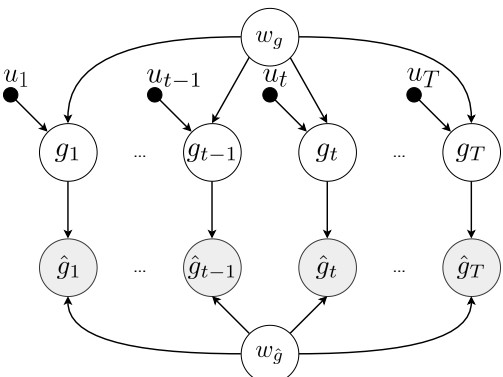

Figure 1: A probabilistic graphical model of our proposed VSGD optimizer. White circles represent latent variables, gray circles correspond to observable variables, filled small black circles denote control variables.

expected value of the posterior distribution over the *true* gradient $g_t$ and plug it into the SGD update rule (Eq. 1). In the following, we present how we can achieve that.

In our considerations, we treat all quantities as random variables. Hence, the *true* gradient $g_t$ is a latent variable, and we observe the *noisy* gradient $\hat{g}_t$. We propose to model $g_t$ and $\hat{g}_t$ using Gaussian distributions with precision variables $w_g$ and $w_{\hat{g}}$, representing the information of systematic and observation noises, respectively. Here, we use "observation noise" to indicate the sampling noise of the observed gradient and "systematic noise" to indicate the introduced error of trying to approximate the real, local area of the gradient surface with a simplified model. Both precision variables are treated as Gamma-distributed latent variables. The advantage of treating precision variables as latent variables instead of hyperparameters is that this way, we can incorporate prior knowledge over the systematic and observation noise as well as having uncertainty over them.

We also introduce a control variate $u_t$ to serve as the prior mean for $g_t$ and is essential for our approach. We define $u_t = h(\hat{g}_{1:t-1})$ as a deterministic message passed from previously observed noisy gradients to time $t$. Later (Eq. 7), we will discuss the specific formulation of the function $h(\cdot)$. In the case of $\hat{g}_t$, we propose using $g_t$ as the mean.

We assume that the gradients across each dimension are independent by modeling them independently. This approach is adopted for computational efficiency. Moving forward, unless specified otherwise, all symbols should be considered as scalars.

For the maximum number of iterations $T$, the joint distribution is then defined as follows (Figure 1):

$$p(w_g, w_{\hat{g}}, g_{1:T}, \hat{g}_{1:T}; u_{1:T}) = p(w_g)p(w_{\hat{g}}) \prod_{t=1}^{T} p(g_t|w_g; u_t)p(\hat{g}_t|g_t, w_{\hat{g}}), \qquad (2)$$

with the following distributions:

$$p(g_t|w_g; u_t) = \mathcal{N}(u_t, w_g^{-1}), \qquad (3)$$

$$p(\hat{g}_t|g_t, w_{\hat{g}}) = \mathcal{N}(g_t, w_{\hat{g}}^{-1}), \qquad (4)$$

$$p(w_g) = \Gamma(\gamma, \gamma), \qquad (5)$$

$$p(w_{\hat{g}}) = \Gamma(\gamma, K_g\gamma), \qquad (6)$$

where the Gamma distributions $\Gamma(\cdot)$ are parameterized by shape and rate. $K_g$ is a hyperparameter designed to reflect our prior belief about the variance ratio between $\hat{g}_i$ and $g_i$, i.e., $K_g = \mathbb{E}\left[w_{\hat{g}}^{-1}\right]/\mathbb{E}\left[w_g^{-1}\right]$. A recommended value for $K_g$ is greater than 1, e.g. $K_g = 30$, suggesting that the majority of observed variance stems from observation noise $w_{\hat{g}}^{-1}$ rather than systematic noise $w_g^{-1}$. $\gamma$ is a shared hyperparameter that indicates the strength of the prior, essentially equating the prior knowledge with $\gamma$ observations. A smaller value for $\gamma$, such as 1e−8, is advised to represent our prior ignorance. The remaining key question is the

choice of the control variate $u_t$ as a way to carry information. In this paper, we aggregate the information over $t$ by setting the prior mean for the current $g_t$ as the posterior mean of the previous ones:

$$u_t = \mathbb{E}_{p(g_t|\hat{g}_{t-1};u_{t-1})}[g_t], \tag{7}$$

where $u_0$ could be set to 0.

An alternative approach to using $u_t$ to summarize the knowledge before $t$ is to instead let $g_t$ explicitly depend on $g_{1:t-1}$. However, incorporating the dependency of $g_t$ on $g_{1:t-1}$ explicitly would lead to a non-scalable, fully connected graphical model. A common alternative solution is to adopt a Markov structure, in which $g_t$ depends only on the most recent $L$ observations $g_{t-L:t-1}$. This would align the model with frameworks similar to Kalman filters or Bayesian filters (Kalman, 1960). However, these models either require known precision terms beforehand, which is often impractical or necessitate online learning of precisions through computationally intensive methods such as nested loops in variational inference or Monte Carlo methods, rendering them non-scalable for training deep neural networks. By introducing $u_t$ and treating it as an additional observation, we can utilize SVI to avoid nested loops and still guarantee convergence to the estimated precisions (if any). As demonstrated in this paper, this strategy proves to be both scalable and efficient.

**Stochastic Variational Inference** Since the model defined by Eqs. 3-6 requires calculating intractable integral over latent variables, to find a good estimate of $g_t$, we aim to approximate the distribution $p(w_g, w_{\hat{g}}, g_{1:T}, \hat{g}_{1:T}; u_{1:T})$ with a variational distribution $q(w_g, w_{\hat{g}}, g_{1:T}; \tau, \phi_{1:T})$, parameterized by the global and local variational parameters $\tau$ and $\{\phi_t\}_{t=1}^T$ respectively. We achieve this by maximizing the evidence lower-bound (ELBO) defined by $p$ and $q$

$$\arg\max_{\tau,\phi_{1:T}} \mathbb{E}_q \left[ \log \frac{p(w_g, w_{\hat{g}}, g_{1:T}, \hat{g}_{1:T}; u_{1:T})}{q(w_g, w_{\hat{g}}, g_{1:T}; \tau, \phi_{1:T})} \right]. \tag{8}$$

We will demonstrate that this objective can be achieved concurrently with the original optimization objective $\theta^* = \arg\max_\theta \mathcal{L}(\theta; \mathcal{D})$, allowing each SGD iteration of $\theta$ to serve simultaneously as an SVI iteration for Eq. 8.

Following SVI (Hoffman et al., 2013), we employ the mean-field approach, assuming the variational posterior is factorized as follows:

$$q(w_g, w_{\hat{g}}, g_{1:T}) = q(w_g)q(w_{\hat{g}}) \prod_{t=1}^T q(g_t). \tag{9}$$

We define the marginal variational distributions as:

$$q(w_g) = \Gamma(a_g, b_g), \tag{10}$$
$$q(w_{\hat{g}}) = \Gamma(a_{\hat{g}}, b_{\hat{g}}), \tag{11}$$
$$q(g_t) = \mathcal{N}(\mu_{t,g}, \sigma_{t,g}^2), \tag{12}$$

for $t = 1 : T$, where the global and local variational parameters are $\tau = (a_g, a_{\hat{g}}, b_g, b_{\hat{g}})^T$ and $\phi_t = (\mu_{t,g}, \sigma_{t,g}^2)^T$, respectively.

We now outline the complete SVI update process to meet the objective set in Eq. 8. For detailed derivations, please see Appendix A.1. Let $\tau_{t-1} = (a_{t-1,g}, a_{t-1,\hat{g}}, b_{t-1,g}, b_{t-1,\hat{g}})^T$ and $\phi_{t-1} = (\mu_{t-1,g}, \sigma_{t-1,g}^2)^T$ represent the global and local variational parameters updated just before the $t$-th SVI iteration, and we set the control variate:

$$u_t = \mu_{t-1,g}, \tag{13}$$

as the variational version of $u_t$ as in Eq. 7, and let $u_0 = 0$. At iteration $t$, we start by randomly drawing a noisy gradient $\hat{g}_t$ with the sampling algorithm of your choice (e.g. using a mini-batch of samples). Then,

conditioned on $\hat{g}_t$, $\tau_{t-1}$, and $u_t$, we update the local parameter $\phi_t$ as follows:

$$\mu_{t,g} = \frac{b_{t-1,\hat{g}}}{b_{t-1,\hat{g}} + b_{t-1g}}\mu_{t-1,g} + \frac{b_{t-1,g}}{b_{t-1,\hat{g}} + b_{t-1g}}\hat{g}_t, \tag{14}$$

$$\sigma_{t,g}^2 = \left(\frac{a_{t-1,g}}{b_{t-1,g}} + \frac{a_{t-1,\hat{g}}}{b_{t-1,\hat{g}}}\right)^{-1}. \tag{15}$$

Note that $\mu_{t,g}$ represents a dynamic weighted average of the previous estimate $\mu_{t-1,g}$ and the noisy observation $\hat{g}_t$. Weights are dynamically determined based on the learned relative precision between Eq. 3 and Eq. 4. $\mu_{t,g}$ then acts as a control variate for the subsequent iteration. Recall that $\frac{a_{t,g}}{b_{t,g}} = \mathbb{E}_{q_t}[w_g]$. Therefore, the variance parameter $\sigma_{t,g}^2$ is equal to the inverse sum of the two precisions characterizing the systematic and observation noise. For an alternative kernel smoothing view on Eq. 14, please refer to Appendix A.2.

Given local parameters $\phi_t$, update equations for the global parameters $\tau_t$ have the form:

$$a_{t,g} = a_{t,\hat{g}} = \gamma + 0.5, \tag{16}$$

$$b_{t,g}' = \gamma + 0.5\left(\sigma_{t,g}^2 + (\mu_{t,g} - \mu_{t-1,g})^2\right), \tag{17}$$

$$b_{t,\hat{g}}' = K_g\gamma + 0.5\left(\sigma_{t,g}^2 + (\mu_{t,g} - \hat{g}_t)^2\right). \tag{18}$$

In Eq. 16, $a_{t,g}$ and $a_{t,\hat{g}}$ remain constant throughout the iterations and depend purely on the prior strength $\gamma$. For the sake of consistency, they are included in the update equations. Thus, the posterior belief of the precision parameters $w_g$ and $w_{\hat{g}}$ are determined purely by the rate parameters $(b_{t,g}', b_{t,\hat{g}}')$. Intuitively, $b_{t,g}'$ and $b_{t,\hat{g}}'$ characterize the amount of systematic noise and observation noise accordingly. In both cases, the prior value and the estimated variance of the true gradient additively contribute to the estimation. The main difference between the two comes in the third term, which depends on how much the gradient mean changed in time for the systematic noise and the difference between the observed gradient and the mean estimate for the observation noise.

Finally, the SVI updates for the global parameters take the form:

$$b_{t,g} = (1 - \rho_{t,1}(t))b_{t-1,g} + \rho_{t,1}(t)b_{t,g}', \tag{19}$$

$$b_{t,\hat{g}} = (1 - \rho_{t,2}(t))b_{t-1,\hat{g}} + \rho_{t,2}(t)b_{t,\hat{g}}'. \tag{20}$$

where $\rho_{t,1}(t)$ and $\rho_{t,2}(t)$ denote the learning rates of SVI across iterations $t$. A typical selection for $\rho$ could be as follows:

$$\rho_{t,1}(t) = t^{-\kappa_1}, \tag{21}$$

$$\rho_{t,2}(t) = t^{-\kappa_2}. \tag{22}$$

In this formulation, $\kappa_1$ and $\kappa_2$ are hyperparameters that influence the behavior of the SVI learning rates. We typically limit $\kappa_1$ and $\kappa_2$ in the range $(0.5, 1]$ to let $\rho_{t,1}$ and $\rho_{t,2}$ meet the Robbins–Monro conditions (Robbins & Monro, 1951), and we select $\kappa_2 > \kappa_1$ to allow for a slower or larger decaying learning rate for $w_g$. For example, setting $\kappa_1 = 0.8$ and $\kappa_2 = 0.9$ allows $w_g$ to adapt more rapidly to new developments compared to $w_{\hat{g}}$.

Note that when using a mini-batch of samples, $\hat{g}_t$ is calculated as the average of the sampled gradients in the batch. While in VSGD it is possible to treat the samples in the mini-batch separately, see Appendix A.3 for details.

Having concluded the $t$-th SVI iteration, we then incorporate the local parameter $\phi_t$ to finalize the $t$-th SGD iteration:

$$\theta_t = \theta_{t-1} - \frac{\eta}{\sqrt{\mu_{t,g}^2 + \sigma_{t,g}^2}}\mu_{t,g}, \tag{23}$$

where $\sqrt{\mu_{t,g}^2 + \sigma_{t,g}^2} = \sqrt{\mathbb{E}[g_t^2]}$ represents an estimation of the local Lipschitz constant. A summary of the steps of the VSGD algorithm can be seen in Algorithm 1.

---

**Algorithm 1** VSGD

---

**Input:** SVI learning rate parameter $\{\kappa_1, \kappa_2\}$, learning rate $\eta$, prior strength $\gamma$, prior variance ratio $K_g$.
**Initialize:**
$\theta_0, a_{0,g} = \gamma; a_{0,\hat{g}} = \gamma; b_{0,g} = \gamma; b_{0,\hat{g}} = K_g\gamma; \mu_{0,g} = 0$
**for** $t = 1$ **to** $T$ **do**
    Compute $\hat{g}_t$ for $\mathcal{L}(\theta; \cdot)$
    $\rho_{t,1} = t^{-\kappa_1}$
    $\rho_{t,2} = t^{-\kappa_2}$
    Update $\sigma_{t,g}^2, \mu_{t,g}$                          $\triangleright$ Eq. 14, 15
    Update $a_{t,g}, a_{t,\hat{g}}$                              $\triangleright$ Eq. 16
    Update $b_{t,g}, b_{t,\hat{g}}$                              $\triangleright$ Eq. 19,20
    Update $\theta_t$                                            $\triangleright$ Eq. 23
**end for**

---

### 3.2 VSGD, Constant VSGD and their connections to other gradient-based deep learning optimizers

Our probabilistic framework offers a way of estimating gradients. A natural question to raise here is whether this framework is related to other 1$^{\text{st}}$ order optimizers used in deep learning. Similarly to well-known methods like ADAM, VSGD maintains a cache of the first and second momenta of the gradient, calculated as weighted moving averages. However, unlike other methods, where these variables remain constant, VSGD adjusts its weights across iterations.

**VSGD and Normalized-SGD**  First, we can show how our VSGD method relates to a version of SGD called NORMALIZED-SGD (Murray et al., 2019) that updates the gradients as follows:

$$\theta_t = \theta_{t-1} - \frac{\eta}{\sqrt{\hat{g}_t^2}}\hat{g}_t. \tag{24}$$

If we take $\gamma \to \infty$, $K_g \to 0$, such that $\lim_{\gamma \to \infty K_g \to 0} \gamma K_g = 0$, then VSGD is approximately equivalent to NORMALIZED-SGD (Murray et al., 2019). We can show it by taking $b_{0,g} = \gamma$, $b_{0,\hat{g}} = K_g\gamma$. Then, for $\gamma \to \infty$ and $K_g \to 0$, Eq. 14 yields the following:

$$\mu_{1,g} = 0 \cdot \frac{K_g\gamma}{K_g\gamma + \gamma} + \hat{g}_1 \frac{\gamma}{K_g\gamma + \gamma} \approx \hat{g}_1. \tag{25}$$

In a subsequent iteration, as $\gamma$ dominates all other values, we have: $a_{t,g} \approx \gamma$, $b'_{t,g} \approx \gamma$, $b'_{t,\hat{g}} \approx K_g\gamma$. As a result, $b_{t,g} \approx \gamma$, $b_{t,\hat{g}} \approx K_g\gamma$. By induction, the variational parameter for $g_t$ at iteration $t$ is the following:

$$\sigma_{t,g}^2 = \frac{K_g\gamma^2}{\gamma(K_g\gamma + \gamma)} = \frac{K_g}{K_g + 1} \approx 0, \tag{26}$$

$$\mu_{t,g} = \mu_{t-1,g} \cdot \frac{K_g\gamma}{K_g\gamma + \gamma} + \hat{g}_t \frac{\gamma}{K_g\gamma + \gamma} \approx \hat{g}_t. \tag{27}$$

As a result, the update rule for $\theta$ is approximately the update rule of NORMALIZE-SGD, namely:

$$\theta_t \approx \theta_{t-1} - \frac{\eta}{\sqrt{\hat{g}_t^2}}\hat{g}_t. \tag{28}$$

**Constant VSGD**  Before we look at further connections between our VSGD and other optimizers, we first introduce a simplified version of VSGD with a constant variance ratio shared between $g_t$ and $\hat{g}_t$. This results in the following model:

$$p(\omega, g_{1:T}, \hat{g}_{1:T}; u_{1:T}) = p(\omega) \prod_{t=1}^{T} p(g_t|\omega; u_t)p(\hat{g}_t|g_t, \omega), \tag{29}$$

with the following distributions:

$$p(g_t|\omega; u_t) = \mathcal{N}(u_t, K_g^{-1}\omega^{-1}), \tag{30}$$

$$p(\hat{g}_t|g_t, \omega) = \mathcal{N}(g_t, \omega^{-1}), \tag{31}$$

$$p(\omega) = \Gamma(\gamma, \gamma). \tag{32}$$

This approach is tantamount to imposing a strong prior on Eq. 2, dictating that the observation noise is the systematic noise scaled by $K_g$. This strong prior affects only the ratio between systematic and observation noises. Regarding the prior knowledge of the observation noise $\omega$ itself, we can maintain an acknowledgment of our ignorance by setting $\gamma$ to a small value such as $10^{-8}$.

Given its characteristic of maintaining a constant variance ratio, we refer to the algorithm as CONSTANT VSGD. Since the derivation of CONSTANT VSGD aligns with that of other variants of VSGD, we omit the detailed derivation here and directly present the update equations.

---

**Algorithm 2** CONSTANT VSGD

**Input:** SVI learning rate parameter $\kappa$, SGD learning rate $\eta$, prior strength $\gamma$, variance ratio $K_g$.
**Initialize:**
$\theta_0, a_{0,\hat{g}} = \gamma; b_{0,\hat{g}} = \gamma; \mu_{0,g} = 0$
**for** $t = 1$ **to** $T$ **do**
    Draw $\hat{g}_t$
    $\rho_t = t^{-\kappa}$
    Update $\sigma_{t,g}^2, \mu_{t,g}$                                                  $\triangleright$ Eq. 33, 34
    Update $a_{t,\hat{g}}, b_{t,\hat{g}}$                                                $\triangleright$ Eq. 35,36
    Update $\theta_t$                                                      $\triangleright$ Eq. 37
**end for**

---

$$\mu_{t,g} = \mu_{t-1,g}\frac{K_g}{K_g + 1} + \hat{g}_t\frac{1}{K_g + 1}, \tag{33}$$

$$\sigma_{t,g}^2 = \frac{1}{K_g + 1}\frac{b_{t-1,\hat{g}}}{a_{t-1,\hat{g}}}, \tag{34}$$

$$a_{t,\hat{g}} = \gamma + 1, \tag{35}$$

$$b_{t,\hat{g}} = (1 - \rho_t)b_{t-1,\hat{g}} + \rho_t\left[\gamma + 0.5\left(\sigma_{t,g}^2 + (\mu_{t,g} - \hat{g}_t)^2\right) + 0.5K_g\left(\sigma_{t,g}^2 + (\mu_{t,g} - \mu_{t-1,g})^2\right)\right], \tag{36}$$

$$\theta_t = \theta_{t-1} - \frac{\eta}{\sqrt{\mu_{t,g}^2 + \sigma_{t,g}^2}}\mu_{t,g}. \tag{37}$$

The complete CONSTANT VSGD algorithm is outlined in Algorithm 2, the updates are notably simpler compared to VSGD, as outlined in Algorithm 1, yet they still offer sufficient flexibility to estimate the uncertainties of the system. Summarizing the discussion above, we enumerate the key distinctions between CONSTANT VSGD and VSGD as follows:

1. In VSGD, two global hidden variables, $w_g$ and $w_{\hat{g}}$, are employed to represent systematic and observation noise, respectively. On the contrary, CONSTANT VSGD utilizes $\omega$ for observation noise and constrains the systematic noise to a fraction $\frac{1}{K_g}$ of the observation noise. Due to these reduced degrees of freedom, all the update equations in CONSTANT VSGD are simplified.

2. In the first momentum update Eq. 14 of VSGD, the weights are determined by the ratio between systematic and observation noise. However, in Eq. 33 of CONSTANT VSGD, these weights are fixed at $\{\frac{K_g}{K_g+1}, \frac{1}{K_g+1}\}$, facilitating a direct comparison with other optimizers such as ADAM.

To sum up, now we know how to simplify VSGD to CONSTANT VSGD. In the following, we will indicate how CONSTANT VSGD is connected to other optimizers, namely: ADAM, SGD with momentum, and AMSGRAD.

### 3.2.1 Constant VSGD and Adam

First, we compare CONSTANT VSGD with one of the strongest and most popular optimizers for DNNs, ADAM (Kingma & Ba, 2015). At the $t$-th iteration, ADAM updates the first two momenta of the gradient through the following procedure:

$$m_t = \beta_1 m_{t-1} + (1 - \beta_1)\hat{g}_t, \tag{38}$$

$$v_t = \beta_2 v_{t-1} + (1 - \beta_2)\hat{g}_t^2, \tag{39}$$

where $\beta_1, \beta_2$ are hyperparameters.

Then, $\theta$ is updated as follows:

$$\theta_t = \theta_{t-1} - \frac{\eta}{\sqrt{\tilde{v}_t}}\tilde{m}_t, \tag{40}$$

where $\tilde{m}_t$ and $\tilde{v}_t$ are the bias-corrected versions of $m_t$ and $v_t$, namely:

$$\tilde{m}_t = \frac{m_t}{1 - \beta_1^t}, \tag{41}$$

$$\tilde{v}_t = \frac{v_t}{1 - \beta_2^t}. \tag{42}$$

Now, the counterparts of $m_t$ and $v_t$ in CONSTANT VSGD, i.e., the first and second momenta:

$$\mathbb{E}[g_t] = \mu_{t,g}, \tag{43}$$

$$\mathbb{E}[g_t^2] = \mu_{t,g}^2 + \sigma_{t,g}^2, \tag{44}$$

where $\mu_{t,g}$ is defined in Eq. 33. If we set $K_g = \frac{\beta_1}{1-\beta_1}$, then the first momentum updates in Eq. 33 and Eq. 38 become identical.

Next, we examine the relationship between Eq. 39 and Eq. 44. Intuitively, $v_t$ (Eq. 39) is an approximation of the expected second momentum of the noisy gradient $\mathbb{E}[\hat{g}_t^2]$. While Eq. 44 attempts to quantify the expected second momentum of the actual gradient $\mathbb{E}[g_t^2]$.

By plugging to Eq. 33 and Eq. 34 to Eq. 44, we obtain the following:

$$\mathbb{E}[g_t^2] = \mu_{t-1,g}^2 \frac{K_g^2}{(K_g + 1)^2} + \hat{g}_t^2 \frac{1}{(K_g + 1)^2} \tag{45}$$

$$+ \frac{2K_g}{(K_g + 1)^2}\mu_{t-1,g}\hat{g}_t + \frac{1}{K_g + 1}\frac{b_{t-1,\hat{g}}}{a_{t-1}}. \tag{46}$$

For clarity, we divide the aforementioned equation into two components: Eq. 45 and Eq. 46. The first component aligns with Eq. 39, as both represent weighted sums of an estimated second momentum and $\hat{g}_t^2$. The distinction arises in the second component, where CONSTANT VSGD introduces two additional factors into the weighted sum: $\mu_{t-1,g}\hat{g}_t$ and $\frac{1}{K_g+1}\frac{b_{t-1,\hat{g}}}{a_{t-1}}$. Specifically, $\mu_{t-1,g}\hat{g}_t$ applies a penalty on Eq.45 when $\mu_{t-1,g}$ and $\hat{g}_t$ have opposing signs. The factor $\frac{1}{K_g+1}\frac{b_{t-1,\hat{g}}}{a_{t-1}}$ represents the $\frac{1}{K_g+1}$ fraction of observation noise learned from the data, while a higher observation noise implies greater uncertainty in $g_t$, leading to a higher expected value $\mathbb{E}[g_t^2]$.

In summary, CONSTANT VSGD described in Algorithm 2 can be regarded as a variant of ADAM. The first momentum update in both algorithms is equivalent when $K_g$ is set to $\frac{\beta_1}{1-\beta_1}$. However, the second momentum update in CONSTANT VSGD incorporates an additional data-informed component that is dynamically adjusted based on the observation noise learned from the data.

### 3.2.2 Constant VSGD and SGD with momentum (SGDM)

The second optimizer to which we compare CONSTANT VSGD is SGD with the momentum term (SGDM). SGDM is quite often used for DNN training due to its simplicity and improved performance compared

to SGD (Liu et al., 2020b). At the $t$-th iteration, SGDM calculates an increment to $\theta_{t-1}$, denoted by $v_t$, expressed as a weighted sum between the previous increment $v_{t-1}$ and the latest noisy gradient $\hat{g}_t$, namely:

$$v_t = \lambda v_{t-1} + \eta \hat{g}_t, \tag{47}$$

$$\theta_t = \theta_{t-1} - v_t, \tag{48}$$

where $\lambda$ is a hyperparameter. After rearranging Eq. 47, we get the following:

$$v_t = \lambda v_{t-1} + \eta \hat{g}_t = (\lambda + \eta) \left[ \frac{\lambda}{\lambda + \eta} v_{t-1} + \frac{\eta}{\lambda + \eta} \hat{g}_t \right]. \tag{49}$$

That is, the increment $v_t$ in SGDM is a weighted average of $v_{t-1}$ and $\hat{g}_t$, the counterpart of $v_t$ in CONSTANT VSGD is $\mu_{t,g}$ updated by Eq. 33. Therefore, by choosing $K_g$ such that $K_g = \frac{\gamma}{\eta}$, the update equations 49 and 33 become identical except for a constant scaling factor.

### 3.2.3   Constant VSGD and AMSGrad

Another optimizer related to CONSTANT VSGD is AMSGRAD (Reddi et al., 2018), a variant of ADAM. AMSGRAD was introduced to address a specific issue of ADAM in estimating the second momentum. To correct the problem, a factor contributing to ADAM's suboptimal generalization performance was introduced. AMSGRAD rectifies it by incorporating the *long-term memory* of the past gradients.

Specifically, the momentum accumulation process in AMSGRAD remains identical to ADAM, as defined in Eq. 38 and Eq. 39. However, for the parameter update step, AMSGRAD differs from AMSGRAD (as in Eq. 40, whether bias-corrected or not) by incorporating the maximum of the past second momentum estimates rather than relying solely on the most recent value, that is:

$$\hat{v}_t = \max(\hat{v}_{t-1}, v_t), \tag{50}$$

$$\theta_t = \theta_{t-1} - \frac{\eta}{\sqrt{\tilde{v}_t}} \tilde{m}_t. \tag{51}$$

Here, the *long-term memory* is carried out by calculating the maximum of all historical second momentum estimations.

Similarly to AMSGRAD, CONSTANT VSGD incorporates a form of long-term memory regarding past gradient information in its second momentum update equations 45 and 46. However, this is not achieved by computing the maximum of historical values. Notably, the factor $\frac{1}{K_g+1} \frac{b_{t-1,\hat{g}}}{a_{t-1}}$ in Eq. 46 represents the expected observation noise, estimated using data up to the $(t-1)$-th iteration. In this expression, while $K_g$ and $a_{t-1}$ remain constant for $t \geq 2$, the variable component is $b_{t-1,\hat{g}}$, which is determined by Eq. 36, for clarity we restate it as:

$$b_{t,\hat{g}} = (1 - \rho_t) b_{t-1,\hat{g}} + \rho_t s, \tag{52}$$

where:

$$s = \gamma + 0.5 \left( \sigma_{t,g}^2 + (\mu_{t,g} - \hat{g}_t)^2 \right) + 0.5 K_g \left( \sigma_{t,g}^2 + (\mu_{t,g} - \mu_{t-1,g})^2 \right). \tag{53}$$

It represents the combined squared systematic and observation noises after considering $\hat{g}_t$. Thus, Eq.52 suggests that $b_{t-1,\hat{g}}$ is a balanced summation of the historical estimate of $b_{\hat{g}}$ and the newly squared disturbances. The forgetting rate decreases over $t$, $\rho_t = t^{-\kappa}$, promoting longer memory retention in successive iterations. In CONSTANT VSGD, the magnitude of the memory retention can be modulated through the hyperparameter $\kappa$.

## 4   Experiments

In this Section, we evaluate the performance of the VSGD compared to other optimizers on high-dimensional optimization problems [1]. The task we focus on in our work is training DNNs to solve classification tasks.

---

[1]Code is available at `github.com/generativeai-tue/vsgd`

### 4.1 Setup

**Data**  We used three benchmark datasets: CIFAR100 (Krizhevsky et al., 2009), TinyImagenet-200 (Deng et al., 2009a)[2], and Imagenet-1k (Deng et al., 2009b). The CIFAR100 dataset contains 60000 small ($32 \times 32$) RGB images labeled into 100 different classes, 50000 images are used for training, and 10000 are left for testing. In the case of TinyImagenet-200, the models are trained on 100000 images from 200 different classes and tested on 10000 images. For a large-scale experiment, we use the Imagenet-1k dataset which contains 1,281,167 images from 1000 classes.

**Architectures**  We evaluate VSGD on various optimization tasks and use three different neural network architectures: VGG (Simonyan & Zisserman, 2015), ResNeXt (Xie et al., 2017), and ConvMixer (Trockman & Kolter, 2023). We keep the model hyperparameters fixed for all the optimizers in our experiments. We provide a detailed description of each architecture in Appendix B.1.

**Hyperparameters**  We conducted a grid search over the following hyperparameters: Learning rate (all optimizers); Weight decay (AdamW, VSGD); Momentum coefficient (SGD). For each set of hyperparameters, we trained the models with three different random seeds and chose the best one based on the validation dataset. The complete set of hyperparameters used in all experiments is reported in Table 3.

Furthermore, we apply the learning rate scheduler, which halves the learning rate after each 10000 training iterations for CIFAR100 and every 20000 iterations for TinyImagenet-200. We train VGG and ConvMixer using batch size 256 for CIFAR100 and batch size 128 for TinyImagenet-200. We use a smaller batch size (128 for CIFAR100 and 64 for TinyImagenet-200) with the ResNeXt architecture to fit training on a single GPU.

### 4.2 Results

In Table 5, we compare the top-1 test accuracy for Adam, AdamW, SGD (with momentum), VSGD, VSGD with weight decay for the different architectures and datasets. The accuracy is averaged across three random seeds. We observe that VSGD almost always converges to a better solution compared to Adam and SGD, outperforming Adam by an average of 2.6% for CIFAR100 and 0.9% for TinyImagenet-200. Additionally, In Figures 2 and 3, we show how the top-1 test accuracy progresses during training. We plot the average values (solid line) as minimum/maximum values (shaded area). We observe that VSGD's convergence is often the same rate or faster compared to AdamW. We present training curves for all models in Appendix B.4.

Table 1: Final Average test accuracy, over three random seeds.

|  | VSGD (w/ L2) | VSGD (w/o L2) | Adam (w/o L2) | AdamW (w/ L2) | SGD (w/ mom) |
|---|---|---|---|---|---|
| CIFAR100 | | | | | |
| VGG16 | **70.1** | 70.0 | 66.8 | 66.6 | 67.9 |
| ConvMixer | **69.8** | 69.1 | 66.5 | 67.0 | 65.4 |
| ResNeXt-18 | **71.4** | 71.2 | 68.2 | 69.7 | 68.5 |
| TinyImagenet-200 | | | | | |
| VGG19 | 51.2 | **52.0** | 47.6 | 49.0 | 50.9 |
| ConvMixer | **53.1** | 52.6 | 51.9 | 52.4 | 52.4 |
| ResNeXt-18 | 48.7 | 47.2 | 48.8 | **48.9** | 47.0 |

Table 2: Average training time on GeForce RTX 2080 Ti (seconds per training iteration) on CIFAR100 dataset.

| Model | | Training Time | |
|---|---|---|---|
| Name | # Params ($\times 10^6$) | Adam | VSGD |
| VGG16 | 14.8 | 0.38 | 0.41 |
| ConvMixer | 0.6 | 0.36 | 0.38 |
| ResNeXt-18 | 11.1 | 0.84 | 0.87 |

**Runtime & Stability**  As can be observed in Algorithm 1, VSGD requires additional operation at each gradient update step compared to the Adam optimizer. However, we did not observe a large computational overhead compared to Adam when training on the GPU. To illustrate this, we provide the average training time (per iteration) in Table 2.

---

[2]We use Tiny Imagenet(Stanford CS231N) provided on `image-net.org`

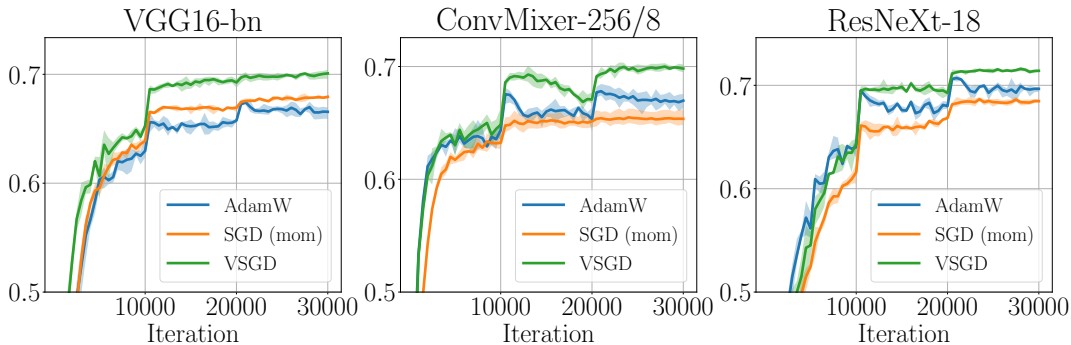

Figure 2: Top-1 test accuracy on CIFAR100 dataset.

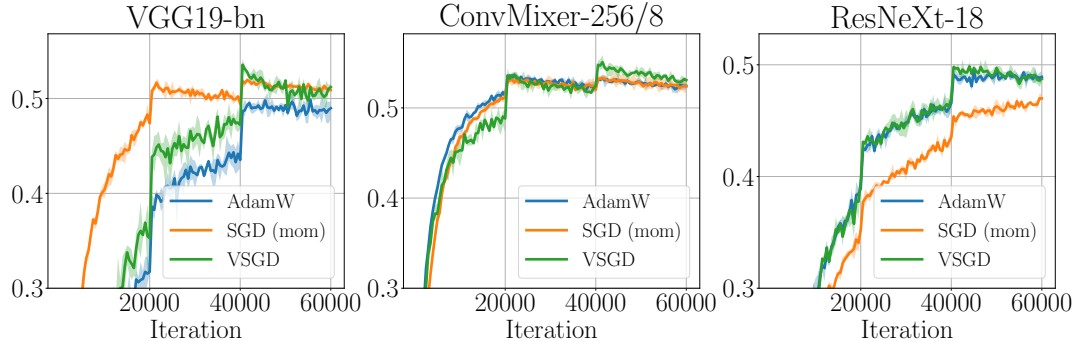

Figure 3: Top-1 test accuracy on TINYIMAGENET-200 dataset.

ADAM is known to perform well on deep learning tasks without the need for extensive hyperparameter tuning. We observed that our approach also demonstrates stable performance without the need to tune the hyperparameters for different architectures and datasets separately. Therefore, we believe that VSGD is generally applicable to various DNN architectures and tasks, as is the case with ADAM.

**Imagenet** In order to evaluate VSGD's performance in overparameterized domains, we performed an experiment on the Imagenet-1k dataset (Deng et al., 2009b) where we trained a ResNet-50 with both VSGD and ADAM. Due to limited computational resources, we did not perform any hyperparameter tuning for this experiment. For both optimizers, we used a learning rate of 0.1 with a ONECYCLELR learning rate scheduler (Smith & Topin, 2019) and a batch size of 256. Same as our previous experiments, we set $\gamma$ to 5e-8 for VSGD. Both models were trained for 50 epochs. The top-1 accuracy of the validation set for VSGD was 72.7%, exceeding ADAM's 69.8% by 2.9%. We also notice slight improvement in convergence speed (Figure 6).

## 5 Conclusion

In this work, we introduced VSGD, a novel optimization framework that combines gradient descent with probabilistic modeling by treating the true gradients as a hidden random variable. This approach allows for a more principled approach to model noise over gradients. Not only does this framework open a new path to optimizing DNNs, but one can establish links with other popular optimization methods by certain specific modeling choices in our method. Lastly, we evaluated VSGD on large-scale image classification tasks using a variety of deep learning architectures, where we demonstrated that VSGD consistently outperformed the baselines and achieved a competitive convergence rate.

In conclusion, we outline two main directions for extending our framework. The first direction is to make stronger dependency assumptions in VSGD. For example, we can model the dependencies between different gradients by introducing covariates between gradients of different parameters in Eq. 3, or considering the second-order momentum of gradients in the VSGD update rules. Second, we advocate for the application of VSGD to a broader spectrum of machine learning challenges beyond classification tasks such as deep generative modeling, representation learning, and reinforcement learning.

## Acknowledgements

This work was carried out on the Dutch national e-infrastructure with the support of SURF Cooperativ. Anna Kuzina is funded by the Hybrid Intelligence Center, a 10-year programme funded by the Dutch Ministry of Education, Culture and Science through the Netherlands Organisation for Scientific Research, `hybrid-intelligence-centre.nl`.

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

# A Derivations

## A.1 Derivation of VSGD

Here, we derive Eq. 14 to Eq. 18 following the SVI technique introduced by (Hoffman et al., 2013). First, we provide a brief overview of the SVI procedure before diving into the details.

In the $t$-th SVI iteration, we realize the following steps:

1. Sample $\hat{g}$ from $\hat{g}_{1:T}$, where the sample is denoted by $\hat{g}_t$.

2. Update the local parameters $\phi_{t,1:T}$ with $\tau_{t-1}$ and $\hat{g}_t$, such that

$$\phi_{t,k} = \underset{\phi_k}{\arg\max} \, \mathbb{E}_{q(w_g, w_{\hat{g}}; \tau_{t-1})} \left[ \log \frac{p(g_k|w_g; u_t)p(\hat{g}_t|g_k, w_{\hat{g}})}{q(g_k; \phi_k)} \right] \qquad k = 1:T, \qquad (54)$$

   where $p(g_k|w_g; u_t)$ and $p(\hat{g}_t|g_k, w_{\hat{g}})$ are defined in Eq. 3 and Eq. 4, respectively. Note that here we use the same $g_t$ for each $k \in 1:T$, as if the same $\hat{g}_t$ is observed $T$ times.

3. Calculate the intermediate global parameters $\tau'_t$ with $\phi_{t,1:T}$, such that:

$$\tau'_t = \underset{\tau}{\arg\max} \, \mathbb{E}_{q(g_{1:T}; \phi_{t,1:T})} \left[ \log \frac{p(w_g)p(w_{\hat{g}}) \prod_{k=1}^{T} p(g_k|w_g; u_t)p(\hat{g}_t|g_k, w_{\hat{g}})}{q(w_g, w_{\hat{g}}; \tau)} \right], \qquad (55)$$

   where $p(w_g)$ and $p(w_{\hat{g}})$ are defined in Eq. 5 and Eq. 6, respectively.

4. Update the global parameters $\tau_t$ using:

$$\tau_t = (1 - \rho)\tau_{t-1} + \rho\tau'_t, \qquad (56)$$

   where $\rho_t = (\rho_{t,1}, \rho_{t,2})^T$ is defined in Eq. 21 and Eq. 22.

5. Pick any $\phi_{t,k} = (\mu_{t,k}, \sigma^2_{t,k})^T \in \phi_{t,1:T}$, and set $u_{t+1} = \mu_{t,k}$, where $u_{t+1}$ serves as the control variate for the next iteration. We can choose any $\phi_{t,k}$ because by the definition (Eq. 54), $\phi_{t,k}$ and $\phi_{t,j}$ are identical for any $i, j \in 1:T$.

Since all $\phi_{t,k}$ are identical for $k = 1:T$, for notational ease, we will use $\phi_t = (\mu_{t,g}, \sigma^2_{t,g})^T$ to represent the unique local parameter updated in the $t$-th SVI iteration. Distributions defined in Eq. 2 are in the conjugate exponential family, therefore, we can derive the closed-form update rule for $\phi_t$ according to Eq. 54:

$$\log q(g_k; \mu_{t,g}, \sigma^2_{t,g}) \equiv \mathbb{E}_{q(w_g, w_{\hat{g}}; \tau_{t-1})} \left[ \log p(g_k|w_g; u_t) + \log p(\hat{g}_t|g_k, w_{\hat{g}}) \right]. \qquad (57)$$

Next, by matching the coefficients of the natural parameters, we get:

$$\mu_{t,g} = u_t \frac{b_{t-1,\hat{g}}}{b_{t-1,\hat{g}} + b_{t-1g}} + \hat{g}_t \frac{b_{t-1,g}}{b_{t-1,\hat{g}} + b_{t-1g}}, \qquad (58)$$

$$\sigma^2_{t,g} = \frac{b_{t-1,\hat{g}} b_{t-1g}}{a_{t-1,g}(b_{t-1,\hat{g}} + b_{t-1g})}, \qquad (59)$$

where Eq. 59 corresponds to Eq. 15. Putting the control variate $u_t = \mu_{t-1,g}$ in Eq. 58 results in Eq. 14. Note that the shape parameters $a_{t-1,g}$ and $a_{t-1,\hat{g}}$ are cancelled out while deriving Eq. 58, reasons being that the shapes are all equal to the same constant over the iterations, as later we will show in Eq. 64.

Similarly for $\tau_t' = (a_{t,g}', a_{t,\hat{g}}', b_{t,g}', b_{t,\hat{g}}')$, according to Eq. 55:

$$\log q(w_g; a_{t,g}', b_{t,g}') \equiv \mathbb{E}_{q(\hat{g}_{1:T};\phi_t)} \left[ \log p(w_g) + \sum_{k=1}^{T} \log p(g_k|u_t, w_g) \right], \tag{60}$$

$$= \mathbb{E}_{q(\hat{g}_{1:T};\phi_t)} \left[ \log p(w_g) + T \log p(g_1|u_t, w_g) \right], \tag{61}$$

$$\log q(w_{\hat{g}}; a_{t,\hat{g}}', b_{t,\hat{g}}') \equiv \mathbb{E}_{q(\hat{g}_{1:T};\phi_t)} \left[ \log p(w_{\hat{g}}) + \sum_{k=1}^{T} \log p(\hat{g}_t|g_k, w_{\hat{g}}) \right], \tag{62}$$

$$= \mathbb{E}_{q(\hat{g}_{1:T};\phi_t)} \left[ \log p(w_{\hat{g}}) + T \log p(\hat{g}_t|g_1, w_{\hat{g}}) \right]. \tag{63}$$

Again, by matching the coefficients of the natural parameters:

$$a_{t,g}' = a_{t,\hat{g}}' = \gamma + 0.5T \tag{64}$$

$$b_{t,g}' = \gamma + 0.5T(\sigma_{t,g}^2 + \mu_{t,g}^2 - 2\mu_{t,g}\mu_{t-1,g} + \mu_{t-1,g}{}^2) \tag{65}$$

$$b_{t,\hat{g}}' = K_g\gamma + 0.5T(\sigma_{t,g}^2 + \mu_{t,g}^2 - 2\mu_{t,g}\hat{g}_t + \hat{g}_t^2) \tag{66}$$

In Eq. 64, $a_{t,g}'$ and $a_{t,\hat{g}}'$ equal the same constant over the iterations. As a result, $a_{t,g}$ and $a_{t,\hat{g}}$ are also equal constants, according to Eq .56.

Eq. 64 to Eq. 66 reveals that during the posterior updates of $w_g$ and $w_{\hat{g}}$, choosing a larger value for $T$ increases the emphasis on the information conveyed by $\hat{g}_t$, consequently diminishing the relative influence of the priors. In VSGD, we have already made the prior strength, $\gamma$, a free parameter, then we can replace $T$ with a constant and still pertain the ability to adjust the strength between prior and new observation through values of $\gamma$.

That means we use a constant value, such as 1, to replace $T$ in Eq. 64, Eq. 65 and Eq. 66, which result in Eq. 17 and Eq. 18. Eq. 19 and Eq. 20 are derived by simply following Eq. 56. Eq. 16 is derived from the fact that Eq. 64 is a constant, and interpolation between two equal constants results in the same constant.

## A.2   A Kernel Smoothing View on VSGD

If we look at VSGD from a kernel smoothing perspective, making use of $u_t$ serves as a cheap approximation to conditioning on all of the historical data. Namely, VSGD seeks to smooth the local gradient function in a similar way with the following kernel smoothing task.

There are two noisy observations of the gradient function at $\theta = \theta_t$. They are $(\theta_t, u_t)$ and $(\theta_t, \hat{g}_t)$. We can define two Gaussian kernels for each of the noisy observations, governed by $w_g^{-1}$ and $w_{\hat{g}}^{-1}$. By denoting the Gaussian kernels $\kappa_1$ and $\kappa_2$, the smoothed gradient value at $\theta_t$ should be as follows:

$$g_t = \frac{\kappa_1(\theta_t, \theta_t)}{\kappa_1(\theta_t, \theta_t) + \kappa_2(\theta_t, \theta_t)} u_t + \frac{\kappa_2(\theta_t, \theta_t)}{\kappa_1(\theta_t, \theta_t) + \kappa_2(\theta_t, \theta_t)} \hat{g}_t. \tag{67}$$

After taking the expectation step of SVI (expect out $w_g$ and $w_{\hat{g}}$), the above equation simplifies to Eq (15).

Note that $u_t$ in VSGD is introduced as a learned summary of the noisy observations before $t$ and we only need to smooth on $u_t$ as a cheap approximation to smoothing on all of the historical data. Similar simplification technique is commonly seen in scalable Gaussian Process solutions, as discussed in Quinonero-Candela & Rasmussen (2005); Liu et al. (2020a). In these references, the learned summaries are called the "inducing points".

If we look at VSGD as an approximation to the local areas of the gradient function, then the precision term $w_g$ can be seen as a smoothness measure of the gradient surface. In VSGD we assume a global $w_g$, which is equivalent to assuming that the smoothness of the whole gradient function is governed by a constant. This assumption is not necessarily ideal, but we still treat it as a global parameter for simplicity and practical reasons.

### A.3 Treating Samples Separately in a Mini-bach

Assuming the size of the mini-batch is $M$, denote the sample gradients $\{\hat{g}_t^{(i)}\}_{i=1:M}$. According to Hoffman et al. (2013), we would need to calculate Eq. 14 $M$ times and average them in Eq. 17 and Eq. 18. Specifically, we would need to replace Eq. 14, Eq. 17 and Eq. 18 with:

$$\mu_{t,g}^{(i)} = \mu_{t-1,g}\frac{b_{t-1,\hat{g}}}{b_{t-1,\hat{g}} + b_{t-1g}} + \hat{g}_t^{(i)}\frac{b_{t-1,g}}{b_{t-1,\hat{g}} + b_{t-1g}} \qquad i = 1:M \tag{68}$$

$$b_{t,g}' = \frac{\sum_{i=1}^{M}\gamma + 0.5\left(\sigma_{t,g}^2 + (\mu_{t,g}^{(i)} - \mu_{t-1,g})^2\right)}{M} \tag{69}$$

$$b_{t,\hat{g}}' = \frac{\sum_{i=1}^{M}K_g\gamma + 0.5\left(\sigma_{t,g}^2 + (\mu_{t,g}^{(i)} - \hat{g}_t^{(i)})^2\right)}{M} \tag{70}$$

And introduce an additional step

$$\mu_{t,g} = \frac{\sum_{i=1}^{M}\mu_{t,g}^{(i)}}{M} \tag{71}$$

before Eq. 23.

Based on the equations presented, the computational cost grows linearly with the batch size $M$. Given the importance of efficiency in contemporary optimization techniques, above equations were omitted from the main paper. Nonetheless, it is conceivable that the increased computational expense could be offset by enhanced convergence rates, a hypothesis reserved for future investigation.

# B Experimental Details

## B.1 Architecture

We have used open-source implementations of all three models.

**VGG**  We use VGG16 and VGG19[3] with batch normalization. We add adaptive average pooling before the final linear layer to make the architecture suitable for both $32 \times 32$ and $64 \times 64$ images.

**ConvMixer**  We use ConvMixer[4] with 256 channels and set the depth to 8.

**ResNeXt**  ResNeXt[5] we set the cardinality (or number of convolution groups) to 8 and the depth to 18 layers. The widen factor is set to 4, resulting in channels being `[64, 256, 512, 1024]`. We add adaptive average pooling before the final linear layer to make the architecture suitable for both $32 \times 32$ and $64 \times 64$ images.

## B.2 Hyperparameters

The full set of hyperparameters used in all experiments is reported in Table 3.

Table 3: Hyperparameter values used in **all** the experiments.

|  | ADAM | VSGD |
|---|---|---|
| $(\beta_1, \beta_2)$ | (0.9, 0.999) | — |
| $(\kappa_1, \kappa_2)$ | — | (0.9, 0.81) |
| $\gamma$ | — | 1e-8 |
| $K_g$ | — | 30 |
| learning rate | {0.001, 0.005, 0.01, 0.02} | |
| weight deacy | {0, 0.01} | |
| momentum coefficient | {0.9, 0.99} | |

## B.3 Data Augmentations

We use data augmentations during training to improve the generalizability of the models. These augmentations were the same for the optimizers and only differ between datasets.

Table 4: Data augmentations.

| CIFAR100 | TINY IMAGENET-200 |
|---|---|
| `RandomCrop(32, padding=4)` | `RandomCrop(64, padding=4)` |
| `RandomHorizontalFlip` | `RandomHorizontalFlip` |
| | `RandomAffine(degrees=45, translate=(0.1, 0.1), scale=(0.9, 1.1))` |
| | `ColorJitter(brightness=0.2, contrast=0.2, saturation=0.2)` |

---

[3]`github.com/alecwangcq/KFAC-Pytorch/blob/master/models/cifar/vgg.py`
[4]`github.com/locuslab/convmixer-cifar10`
[5]`github.com/prlz77/ResNeXt.pytorch`

## B.4 Training Loss

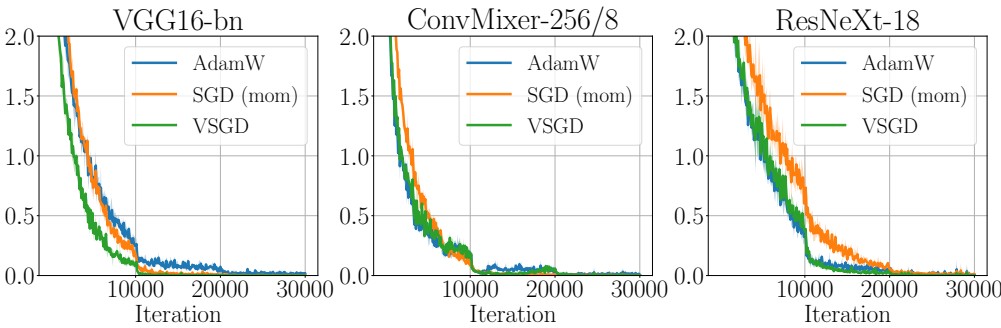

Figure 4: Training loss on CIFAR100 dataset.

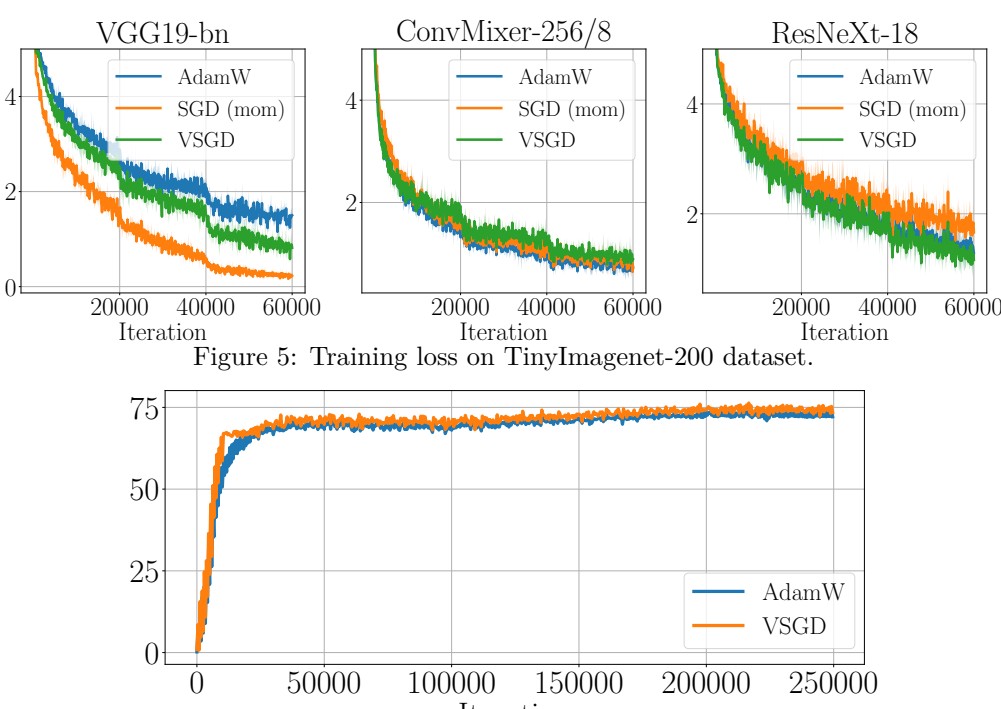

Figure 5: Training loss on TinyImagenet-200 dataset.

Figure 6: Top-1 Accuracy on the Imagenet dataset trained with ResNet-50.

Table 5: Final Average train loss, over three random seeds.

|  | VSGD (w/ L2) | VSGD (w/o L2) | ADAM (w/o L2) | ADAMW (w/ L2) | SGD (w/ mom) |
|---|---|---|---|---|---|
| | CIFAR100 | | | | |
| VGG16 | **0.0016** | 0.0022 | 0.0284 | 0.0157 | 0.0018 |
| CONVMIXER | **0.0019** | 0.0022 | 0.0094 | 0.0094 | 0.0027 |
| RESNEXT-18 | 0.0027 | **0.0015** | 0.0081 | 0.0088 | 0.0151 |
| | TINYIMAGENET-200 | | | | |
| VGG19 | 0.8783 | **0.1900** | 1.2528 | 1.4722 | 0.1954 |
| CONVMIXER | 0.8823 | 0.8513 | **0.7347** | 0.6882 | 0.7581 |
| RESNEXT-18 | 1.3233 | **1.2680** | 1.4096 | 1.4089 | 1.8158 |

## C  Hyperparameter Sensitivity Analysis

In this Section, we investigate the shape parameter $\gamma$'s sensitivity to VSGD's performance. Higher $\gamma$ implies a stronger weight on the prior while a lower $\gamma$ implies a stronger weight on the current observation. We train VSGD on a wide range of $\gamma$ values on CIFAR100 using the CONVMIXER architecture. The results can be observed in Figures 7, 8 and Table 6. Inspecting Figure 7 (Right) more closely, we observe that while there is a statistically significant difference between accuracies obtained by different $\gamma$s, they all perform relatively well as long as $\gamma$ is not too large.

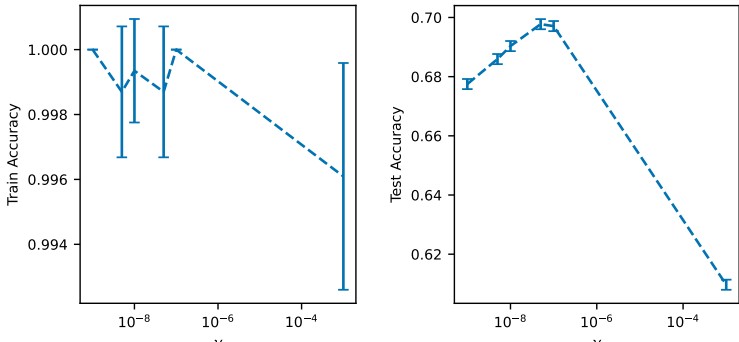

Figure 7: Analysis of the hyperparameter $\gamma$'sensitivity on the VSGD's performance in terms of accuracy.

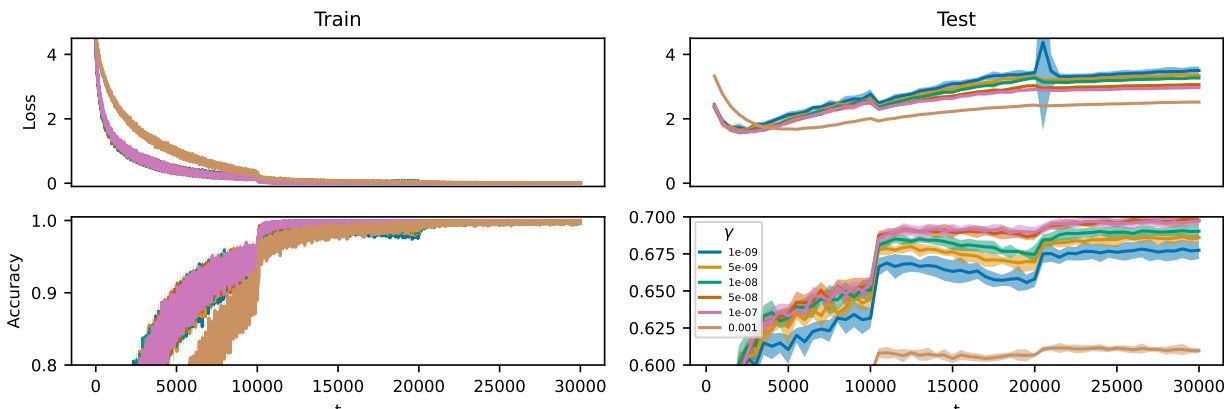

Figure 8: CIFAR100 learning curves trained with CONVMIXER architecture for different $\gamma$ values.

| $\gamma$ | Accuracy |
|---|---|
| 1e-9 | 67.75 |
| 5e-9 | 68.59 |
| 1e-8 | 69.03 |
| 5e-8 | 69.77 |
| 1e-7 | 69.71 |
| 1e-3 | 60.97 |

Table 6: Test accuracy on CIFAR100 trained with CONVMIXER for different $\gamma$ values.

## D  Constant VSGD

In this Section, we offer a comparison between VSGD and CONSTANT VSGD. From the modeling point of view, VSGD has more free parameters that allow our model to learn more complicated distributions. While

Constant VSGD imposes a constraint on VSGD, which is less flexible but may lead to better out-of-sample performance. To investigate this, we ran additional experiments on CIFAR100 with ConvMixer architecture. The final accuracy for VSGD was $69.04 \pm 0.4$, while the accuracy for Constant VSGD was $68.49 \pm 0.19$. The learning curves can be observed in Figure 9.

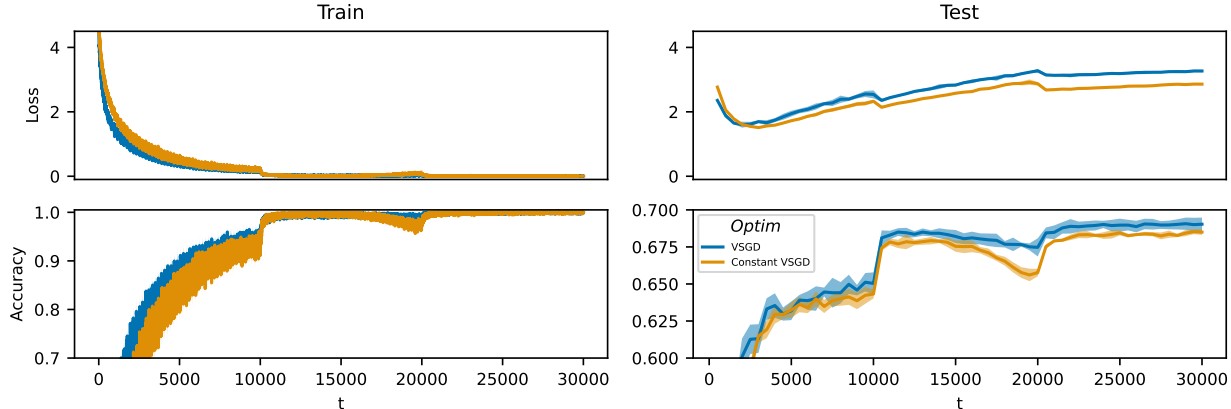

Figure 9: Comparison of Constant-VSGD and VSGD learning curves trained on CIFAR100 with ConvMixer architecture. The final accuracy for VSGD is $69.04 \pm 0.4$, while the accuracy for Constant-VSGD is $68.49 \pm 0.19$.

