# OpenReview forum: "Variational Stochastic Gradient Descent for Deep Neural Networks"
_TMLR — Accepted by TMLR_

### Review · Reviewer_moMe · 2024-10-01

**Summary Of Contributions:**

The authors consider a probabilistic model for describing the relationship between noisy and true gradients during an optimization process. By introducing various priors and parametric approximations, they derive a family of new optimization algorithms. They compare this family of algorithms mathematically to several established accelerated descent algorithms, showing an equivalence with a version of normalized gradient descent in a degenerate prior limit, and highlighting both similarities and differences between the optimization rules utilized in ADAM and AMSGRAD. The authors then compare the performance of their algorithm with ADAM and SGD on some test data sets, showing favorable (modest) performance improvements without a significant increase in computation time.

**Audience:**

Yes

**Claims And Evidence:**

Yes

**Requested Changes:**

Spell check the paper.

**Strengths And Weaknesses:**

Strengths of the submission:
- The main ideas are pretty clear, and the probabilistic framework does introduce a degree of flexibility which could be useful in some problems.
- The effort to compare and contrast with other accelerated descent algorithms is appreciated, and strengthens the work.
- I appreciated that the number of added hyperparameters remained small, making comparison with other algorithms reasonable.
- The numerical experiments, while modest, still do tell a favorable story.

Weaknesses:
- In optimization I think many of the assumptions (especially about independence, for example of the gradient across coordinates) are not always realistic. I understand that this makes it tractable, and in some contexts it's useful, but I'm not sure if the Bayesian framework is really the "right" way to model an optimization process.
- The notation (which doesn't differentiate between vectors and scalars) makes some parts of the paper hard to read and evaluate. I'm assuming the authors mean that the square/square root of a vector is meant in a pointwise sense? Depending on how this is intended some of the comparison with normalized descent may need to be modified.

---

> ### Author Response · Authors · 2025-02-10
> **Response**
>
> We thank the reviewer for thoughtful feedback!
>
> **Many of the assumptions are not always realistic**
>
> We appreciate the reviewer’s recognition of the trade-off between realism and tractability in our assumptions. Regarding the use of the Bayesian framework, our intention is to enable greater adaptiveness in the gradient algorithm. For instance, while Adam adjusts gradients in response to new noisy observations, its adaptation parameters (e.g., beta and learning rates) are fixed at the outset. In contrast, the Bayesian framework added another layer of flexibility that permits these adaptation rules to evolve over time. A stronger prior will steer the method to behave similarly to established adaptive optimizers like Adam, whereas a weaker prior allows the adaptation rules to be more data-driven.
>
>
>
> **The notation doesn't differentiate between vectors and scalars**
>
> As mentioned in the two paragraphs above equation 2: “*We assume that the gradients across each dimension are independent by modeling them independently. This approach is adopted for computational efficiency. Moving forward, unless specified otherwise, **all symbols should be considered as scalars***”. It also indicates that all of the algorithms listed in the paper are elementwise.
>
>
> **Spell check the paper.**
>
> Thank you for the suggestion. We did our best to find and correct all the typos.

---

### Review · Reviewer_FMuq · 2024-11-16

**Summary Of Contributions:**

> Summary

The paper proposes to use a probabilistic model over the gradients using variational inference, named Variational Stochastic Gradient Descent (VSGD). This yields a general and novel way of optimizing deep neural networks. Relationships to other popular optimizers, such as Adam are discussed. The paper presents empirical evicence in favor of using VSGD.

**Audience:**

Yes

**Broader Impact Concerns:**

No ethical implications.

**Claims And Evidence:**

Yes

**Requested Changes:**

See weaknesses and questions above.

The most critical requested change mentioned would be providing training losses.

**Strengths And Weaknesses:**

> Strengths

- The method proposes an elegant probabilistic model over gradient, while remaining cognizant of the memory and computational constraints that are dealt with when optimizing deep neural network. As a result, the method needs to make some strong (independence) assumptions to remain tractable (see weaknesses), but the fact that it is very clear what the assumptions are should be regarded as positive.
- The paper compares with other well-known optimizers, such as Adam, and gives an in-depth comparison under which settings individual update rules are equivalent or in what way they are different.
- Although experiments on a wider range are needed to make more general claims, the model outperforms strong existing baselines such as Adam on a range of well-known vision models.

> Weaknesses and questions

Q1. Understanding the way in which this method is probabilistic.

The abstract states that the method may lead to improved "modeling uncertainties". Although it is not explicitly mentioned, the reader may think this means the method may lead to better uncertainty estimates in the output. If this is meant by the authors, I do not understand why this would be necessarily be the case since, although a probabilistic model over the gradients is employed, the method ultimately still targets the empirical risk (as described in Sec. 2) and thus no form of calibration. The paper also mentions that the method is "orthogonal" to [Khan et al., 2018; Osawa et al.] as VSGD assumes a Bayesian model over the gradients and not applied to a Bayesian inference problem. To my understanding, the proposed method is Bayesian about the gradients, but not the model itself. I am not completely satisfied by the "orthogonal" in this sentence and it would be helpful to better understand the distinction between these lines of work. Could the authors elaborate?

Q2. Gamma parameter

I am a bit concerned about the sensitive analysis of the gamma parameters. Figure 8 seems to suggest that a monotonically lower test loss for lower values of gamma. To get a complete picture, it would be nice to include train losses in Table 5 and Figure 7. Have you tried gamma=0, and/or is this case equivalent to an existing optimization algorithm?

Q3. Train loss missing

It would be helpful to also provide final training losses (the thing we are optimizing!). To me, it is important to distinghuish possible regularizing effects from being a good optimizer. Could training losses be provided? (e.g. train losses in Table 1, Table 5, Figure 7)

Possibly related to this, it seems that lower values of gamma always result in a better loss, but not necesarily test accuracy. Do the authors have an idea what could cause this?

Q4. Memory

It seems that due to the strong independence assumptions, there is no need to perform any inverses (no correlations between weights or need to do filtering etc..). Table 2 is helpful to see that this indeed leads to negligible increase in runtime.

In addition to runtime, it would be helpful to provide an overview of total memory (e.g. as a function of parameters) that is required to maintain all (hyper)-parameters, as well as existing methods such as Adam that are discussed.

Experimental validation

The experimental validation is limited to computer vision tasks. To assess whether the method could be generally better than optimizers such as Adam a wider set of architectures and tasks would be regarded. Yet, the paper is clear about this and does not seem to claim the method to be generally better than Adam/etc. Therefore, I think the experiments to be sufficient for acceptance in PMLR.


> Conclusion

The work proposes an interesting novel method to improve optimization of deep neural networks while incorporating uncertainty over gradient information. The work demonstrates improved performance and relates the work to other existing methods.

---

> ### Author Response · Authors · 2025-02-10
> **Response**
>
> We thank the reviewer for thoughtful feedback!
>
>
> **Q1. Understanding the way in which this method is probabilistic.**
>
> Thank you for the insightful comments. To clarify:
>
> 1. On “modeling uncertainties”:
>     In the abstract, our goal is to estimate the hidden, true gradients by explicitly modeling the uncertainty in the observed gradients. This is analogous to a probabilistic filtering problem—much like inferring the true position of an aircraft from noisy radar data. Thus, “modeling uncertainties” here refers to capturing the uncertainty in the gradient estimates, not necessarily providing calibrated uncertainty estimates for the final model outputs.We will adjust the abstract to add more clarity.
> 2. On “orthogonal” versus Bayesian approaches:
>     You are correct that our approach is Bayesian with respect to the gradients, whereas the work by Khan et al. (2018) is about applying Bayesian inference to the entire model. We used “orthogonal” to emphasize that these methods approach uncertainty estimation from different perspectives. We will revise the manuscript to better articulate this distinction and to adjust the phrasing around “Bayesian inference problem” for clarity.
>
> We appreciate your suggestions and will update the manuscript accordingly.
>
>
>
> **Q2.1 Gamma parameter**
>
> Loss landscape of the deep neural networks is known to have many local minimas. In this sensitivity analysis we report training loss on the top-left plot in Figure 8. We observe that all the models converged to nearly zero training loss region. However, from the test loss and accuracy we see that these are local minimas with different performance on the held-out-dataset.
>
>
> **Q2.2.  Have you tried gamma=0, and/or is this case equivalent to an existing optimization algorithm**
>
> Gamma is a hyperparameter controlling the shape and rate of the gamma distribution which is positive by definition, therefore, we are not able to set it to zero.
>
>
>
> **Q3. Train loss missing**
>
> We report the training loss curves for the experiments in Table 1 in Figures 4 and 5 and the training loss for the hyperparameter sensitivity analysis is reported in Figure 8. However, in deep learning we are usually interested in test loss / accuracy as it shows how well the model generalizes to unseen data. If the model is powerful enough the training loss will often be zero (e.g. Figure 4), but it may correspond to different local minima, which can be observed by different test loss and accuracy values (Table 1).
>
> We have updated the appendix B.4 to additionally report final training loss.
>
>
> **it seems that lower values of gamma always result in a better loss, but not necessarily test accuracy. Do the authors have an idea what could cause this?**
>
> This is indeed an interesting observation. We observe that bigger values of gamma result in better test loss (we assume that is a type in the question). For most the experiments better test loss also corresponds to higher accuracy, except for a very large gamma value, where we observe better test loss but worse test accuracy. This can indicate that the model makes a lot of mistakes but they are all small in the continuous loss space.
>
>
> **Q4. Memory**
>
> VSGD does not have an additional memory overhead compared to Adam. Adam needs to store the first and the second momentum estimate for each trainable parameter, VSGD stores gradient’s mean and variance estimate for each trainable parameter. All other parameters are scalar values (single scalar for all the model parameters), which are negligible in the memory footprint of the optimizer.

---

### Review · Reviewer_DBpR · 2025-01-27

**Summary Of Contributions:**

- The paper introduces Variational Stochastic Gradient Descent (VSGD), a novel probabilistic optimizer for deep neural networks that models gradients as random variables and uses stochastic variational inference to improve gradient estimation and handle noise effectively.
- VSGD is shown to generalize and unify existing optimizers like Adam and Normalized-SGD by offering a probabilistic framework, adapting weights dynamically, and incorporating noise models.
- Experimental results on image classification tasks using CIFAR100, TinyImagenet-200, and Imagenet-1k datasets demonstrate that VSGD achieves higher accuracy and faster convergence compared to Adam and SGD.
- The computational overhead of VSGD is minimal, and the optimizer exhibits stable performance across different architectures without requiring extensive hyperparameter tuning.

**Audience:**

Yes

**Broader Impact Concerns:**

Nothing

**Claims And Evidence:**

Yes

**Requested Changes:**

1. Simplify the modeling framework or, at least, convince me by motivating the necessity of your assumptions more clearly.

2. Correct the variational objective in Equation (8) and ensure the ELBO derivation aligns with the actual model structure.

3. Expand the experimental campaign to include tasks beyond image classification, such as generative models (e.g., VAEs or GANs), text classification with small transformer models, and reinforcement learning tasks. These additions would better demonstrate VSGD’s generalizability and practical utility.

4. Include a sensitivity analysis for other key hyperparameters, and add an empirical study to analyze the impact of batch size on VSGD’s performance and investigate how hyperparameters such as learning rate, $K_g$, and $\gamma$ should be adjusted for larger batch sizes. This would provide practical guidance for users in scaling the optimizer to different hardware configurations.

### Minor Changes

In the second paragraph of the introduction (2nd sentence), the paper implies that momentum combined with SGD was proposed by Sutskever et al. (2013). However, momentum was well-known and discussed in the neural network community long before, as early as Rumelhart et al. (1986) at least. Please revise this statement.

**Strengths And Weaknesses:**

Sorry for the formatting below, but Openreview is refusing to correctly render the equations. I give you directly the raw markdown (it works in vscode).

```md
### Strengths

#### Interesting approach to apply Bayesian principles to gradient estimation

The paper introduces Variational Stochastic Gradient Descent (VSGD), an interesting optimization framework that incorporates Bayesian principles into gradient estimation. By treating true gradients as latent variables and noisy gradients as observed variables, VSGD models the dependencies between them using Gaussian distributions with learnable noise parameters. The approach leverages stochastic variational inference (SVI) to estimate the posterior distributions of the true gradients, allowing for more robust and noise-resilient updates. This Bayesian framework also facilitates the incorporation of prior knowledge about gradient noise variance, which can improve the efficiency and adaptability of the optimization process.

#### Empirical results suggest good performance, at least on the CIFAR100 benchmark

The experimental results highlight VSGD’s good empirical performance, particularly on the CIFAR100 benchmark. The optimizer outperformed Adam and SGD in terms of top-1 test accuracy across various neural network architectures, including VGG16, ResNeXt, and ConvMixer. For instance, VSGD achieved an average improvement of 2.6% over Adam and a 3.2% improvement over SGD on CIFAR100. These results indicate that the proposed approach can consistently achieve lower generalization errors and maintain competitive or superior convergence rates. While the results are promising, it is worth noting that the gains on other datasets like TinyImagenet-200 and Imagenet-1k, though positive, are comparatively smaller, suggesting further evaluation are necessary to generalize the findings fully.

### Weaknesses

#### Modeling assumption is too convoluted and unnecessary

While the Bayesian framework effectively models $g_t$ as the mean of the Gaussian distribution over $\hat{g}_t$, the introduction of separate latent precision variables ($w_g$ and $w_{\hat{g}}$) for systematic and observation noise adds unnecessary complexity. Specifically, the current approach relies on a factorized joint distribution (Equation 2):

$$
p(w_g, w_{\hat{g}}, g_{1:T}, \hat{g}_{1:T}; u_{1:T}) = p(w_g)p(w_{\hat{g}}) \prod_{t=1}^T p(g_t | w_g; u_t)p(\hat{g}_t | g_t, w_{\hat{g}}).
$$

However, in cases where $p(g_t) = \mathcal{N}(u_t, w_g^{-1})$ and $p(\hat{g}_t | g_t) = \mathcal{N}(g_t, w_{\hat{g}}^{-1})$, the marginal distribution $p(\hat{g}_t)$ can be computed directly:

$$
p(\hat{g}_t) = \mathcal{N}(u_t, w_g^{-1} + w_{\hat{g}}^{-1}),
$$

avoiding the need to explicitly model the latent $g_t$. In the context of the paper, $\hat{g}_t$ corresponds to the noisy gradient, $g_t$ to the true gradient, and $u_t$ to the prior mean of $g_t$. This raises some questions about the overall justification for the modeling choices.

On the other hand, this simplification relies on the assumption that the true gradient $g_t$ and the noisy gradient $\hat{g}_t$ are Gaussian distributed. The explicit Gaussian assumptions _before introducing any variational framework_ for both $g_t$ and $\hat{g}_t$ do not necessarily align with the practical distribution of gradients in real-world deep learning problems, which are often non-Gaussian and exhibit heavy-tailed behavior.

#### The variational objective in Equation (8) is flawed

The evidence lower bound (ELBO) defined in Equation (8) aims to approximate the joint distribution over four sets of random variables $(w_g, w_{\hat{g}}, g_{1:T}, \hat{g}_{1:T})$ using a variational distribution that only includes three random variables $(w_g, w_{\hat{g}}, g_{1:T})$. Unless I'm mistaken this discrepancy in dimensionality makes the objective technically invalid as an ELBO since it does not properly bound the marginal log-likelihood of the observed gradients.

However, the redundancy between $g_{1:T}$ and $\hat{g}_{1:T}$, as noted above, might explain why things work out in the end. Specifically, $\hat{g}_{1:T}$ can be expressed as noisy observations of $g_{1:T}$ via Equation (4), and the posterior distribution over $g_{1:T}$ implicitly determines the distribution of $\hat{g}_{1:T}$.

Nonetheless, the inconsistency in the formulation highlights a lack of clarity in the probabilistic modeling, that needs to be fixed.

#### Limited experimental campaign

The experimental evaluation primarily focuses only on image classification tasks. While these experiments demonstrate that VSGD can perform well against Adam and SGD in this context, the narrow scope of the evaluation limits the generalizability of the findings.

Optimization methods like VSGD would be intended to be broadly applicable across a variety of machine learning tasks, yet the paper does not explore other important domains. For instance:

- Generative Models: VSGD could be tested on Variational Autoencoders (VAEs) to assess its ability to optimize latent variable models where gradient dynamics can differ significantly from classification tasks.
- Text Classification: Transformers and/or RNNs for text classification would provide insights into how VSGD performs in NLP tasks, where optimization challenges such as vanishing gradients and long-term dependencies are prevalent.
- Reinforcement Learning: Testing on reinforcement learning tasks with a simple policy gradient method could further validate VSGD’s capacity to manage possibly non-stationary rewards and high-variance gradients.
```

---

> ### Author Response · Authors · 2025-02-10
> **Response**
>
> We thank the reviewer for thoughtful feedback!
>
>
> **Simplify the modeling framework**
>
> We appreciate your insight regarding the marginal distribution. However, if all $g_t$ variables are integrated out, the expected value of
>  $ \hat g_t$  reduces to $u_t$. By the definition of $u_t$, we have
> $
> E(\hat g_t) = u_t = E(\hat g_{t-1}) = u_{t-1} = \cdots = u_0,
> $
> which implies that $\hat g_t$ behaves as if it was generated from a static model. In contrast, incorporating $g_t$ permits the expectation of $\hat g_t$ to evolve over time, thereby enabling the model to adapt dynamically to new observations.
>
>
> **Correct the variational objective in Equation (8)**
>
> We treat noisy gradients $\hat g_t$ as observed random variables, therefore, we do not need to train a variational distribution over it and ELBO in Eq.8 is defined as a function of $\hat g_t$.
>
>
> **Expand the experimental campaign**
>
> We sincerely appreciate the reviewer’s suggestion and agree that additional experiments on LLMs, DGMs, or RL could provide further insights. However, we respectfully believe that the experiments presented in our paper sufficiently demonstrate the effectiveness of our approach. In optimization research, experiments typically vary along multiple axes, including the task/objective, dataset, and architecture. In our work, we have systematically varied both the datasets and architectures to assess the robustness of our optimizer. This aligns with standard practices in optimization research, where it is common to focus on a single task—often classification—while varying other factors to ensure broad applicability (e.g., [1,2]). Given this precedent, we believe our experimental setup is well-justified and provides meaningful evidence of our method’s effectiveness
>
> [1] Zou, Fangyu, et al. "A sufficient condition for convergences of adam and rmsprop." Proceedings of the IEEE/CVF Conference on computer vision and pattern recognition. 2019.
>
> [2] Smith, Leslie N., and Nicholay Topin. "Super-convergence: Very fast training of neural networks using large learning rates." Artificial intelligence and machine learning for multi-domain operations applications. Vol. 11006. SPIE, 2019.
>
> **Revise citation in paragraph 2 of the introduction**
>
> Thank you for pointing this out. We have clarified this sentence.

---

### Public Comment · ~Laurence_Aitchison1 · 2024-09-02
**Suggested reference**

Might be worth citing:

https://arxiv.org/abs/1807.07540

as an alternative attempt at putting optimisation algorithms within a Bayesian framework.

---

### Decision · Action_Editor_xy5r · 2025-04-07

**Recommendation:** Accept as is

**Comment:**

The three reviewers were all knowledgeable and the authors engaged in appropriate response discussions. While two reviewers were positive, one reviewer raised concerns about the modelling assumptions, the ELBO derivations, and the significance of results. The authors responded, but the issues were only partially concluded between the authors and reviewer. Looking at the issue, I am leaning on the positive reviewer's and authors side.

**Audience:**

The domain of probabilistic optimisation of neural networks is well-established, and this work makes non-trivial contributions to it. There is surely interested audience for this work. All reviewers are positive.

**Claims And Evidence:**

The paper proposes to model the uncertainty of stochastic gradients using variational principles, presents a such gradient descent method, draws connections to existing gradient descent methods, and shows empirical improvements. The paper's claims are appropriately defined in the introduction, and have sufficient evidence. All reviewers are positive on this.